# Covid-19 vaccine hesitancy and its predictors among diabetic patients on follow-up at public hospitals in Nekemte Town, Western Ethiopia

Aberash Olani Kuta[1,2], Nagasa Dida[1,3]*

1 Department of Public Health, Faculty of Health Science and Technology, Rift Valley University, Ambo Campus, Ambo, Ethiopia, 2 Student Services Unit, Wollega University, Nekemte, Ethiopia, 3 Department of Public Health, Medicine and Health Science College, Ambo University, Ambo, Ethiopia

* aberashkuta@gmail.com

## Abstract

### Background

Understanding and addressing the concerns of vaccine-hesitant individuals, including those with chronic diseases, is key to increasing vaccine acceptance and uptake. However, in Ethiopia, there is limited evidence on the COVID-19 vaccine hesitancy and predictor variables among diabetic patients. Hence, the study aimed to assess Covid-19 Vaccine Hesitancy and Predictor variables among Diabetic Patients on Follow-Up at Public Hospitals in Nekemte Town, Western Ethiopia.

### Method

Facility based cross sectional study was conducted among 422 diabetic patients attending public hospitals at Nekemte Town, Western Ethiopia between January, to February, 2023. Study participants were recruited by systematic random sampling. The data were collected interviewee administered pre-tested structured survey questioner. The collected data were entered and cleaned using Epi-Data software 4.6 version. The cleaned data were analyzed using SPSS. 25.0 Statical software. Descriptive statistics like frequency, mean and percentage, and binary logistic regression was applied to identify independent predictors of Covid-19 vaccine hesitancy and association between variables were declared at p-value of 0.05.

### Result

The overall magnitude of COVID-19 vaccine hesitancy was 15.2% (95% CI: 11.6–18.7). The top three listed reasons for the COVID-19 vaccine hesitancy were: negative information about the vaccine (32.90%), lack of enough information (21.80%), and vaccine safety concern (19.40%). The hesitancy of the COVID-19 vaccination uptake among diabetes patients was independently influenced by age between 40–49 (Adjusted Odd Ratio [AOR] = 4.52 (1.04–19.66)), having vaccine awareness (AOR = 0.029(0.001–0.86)), having a great deal of trust on vaccine development (AOR = 0.028(0.002–0.52)), and a fear amount trust (AOR

**Data Availability Statement:** All relevant data are within the paper and its Supporting Information files.

**Funding:** The author(s) received no specific funding for this work.

**Competing interests:** The authors have declared that no competing interests exist.

= 0.05(0.003–0.79)) on the vaccine preparation, vaccinated for COVID-19 (AOR = 0.13 (0.04–0.51)), perceived exposure to COVID-19 infection after having the vaccine as strongly agree/agree (AOR = 0.03(0.01–0.17))and neither agree nor disagree (AOR = 0.07(0.02–0.30)).

## Conclusion

COVID-19 vaccine hesitancy among diabetic patients was relatively low. The identified independent predictors were age, vaccine awareness, COVID-19 vaccination history, awareness on vaccine preparation and exposure status to COVID-19 infection. The relevant agency should focus on efforts to translating these high levels of vaccine acceptance into actual uptake, through targeting identifying predictor variables and vaccine availability for a high-risk diabetes patient.

## Introduction

COVID 19 contributed significant public and economic problem worldwide. The existing control measures and vaccine did not able to stop disease transmission, hospitalization and death tone associated with this disease. As of 2 July 2023, the pandemic had caused more than 767 million cases and 6.9 million deaths globally, making it one of the deadliest infectious diseases in the history As a result, Ethiopia has launched COVID-19 vaccination campaign on 16 November 2021, targeting people aged 12 years and above [1].

Vaccines are one of the greatest accomplishments in the history of public health. They have indisputably contributed to a decline in sickness and death from numerous infectious diseases. However, the vaccines hesitancy is determined to be a major threat to the impact of vaccination in the prevention of infection, hospitalization and mortality from the COVID-19 [2, 3]. The existing studies documented the COVID-19 vaccine uptake is suboptimal among people with chronic medical conditions including DM who are at increased risk of complications and mortality associated with SARS-CoV-2 infection. Some recent studies have also reported the magnitude of vaccine hesitancy varying from 3.0% to 76.4%, indicating variabilities across different countries and between different time points [2–4]. Moreover, myths and conspiracy theories on vaccinations have been spreading and can easily be accepted and affect vaccine acceptance. This may cause people to be reluctant towards vaccination, which has been demonstrated by a study in Nigeria by a low vaccine acceptability rate [5]. Furthermore, the WHO listed vaccine hesitancy as one of the ten global threats to public health [6].

A vaccine hesitancy is caused by complex, context specific factors that vary across time, place, and different vaccines, and is influenced by issues such as complacency, efficacy, safety, convenience, price, confidence, and sociodemographic contexts [5, 7]. It is also related to misinformation and conspiracy theories which are often spread online, including through social [5, 7].

In African countries the tendency toward acceptance of COVID-19 vaccine reaches from 81.6% in South Africa to 65.2% in Nigeria [4, 8]). In Ethiopia, the vaccine hesitancy range between 19.1%- 60.3%% were documented in the studies conducted in different part of the country. Such variation in willingness to accept a COVID-19 vaccine may result in difference in vaccine coverage and delay global control of the pandemic [9–11].

Therefore, it is imperative to understand vaccine hesitancy and its predictors among high-risk population like DM patient to design strategies to overcome the vaccine hesitancy. First,

unravelling the specific fears and doubts of the DM patient with a medical condition that increases the risk of infection and complications from COVID-19 can be helpful. This understanding can then assist government and other concerned officials in designing policies and strategies to adequately address the problem; Diabetic Mellitus diseased year–the period for which the patients diagnosed for diabetic mellitus; overall health–When patients are asked about their "perceived condition" of their "overall health status," they are expressing their personal view of their general health and well-being.

## Methods

### Study design, and setting

A facility-based cross-sectional study was conducted between January to February, 2023, at public hospitals providing diabetic follow-up services in Nekemte Town, Western Ethiopia. Nekemte town is located in the Western part of the Oromia region 331 km away from country's capital, Addis Ababa, and astronomical location at 9˚ 04' North Latitude and 36˚ 30' East Longitude. The data from Nekemte Town Health Office showed that the town has four public health institutions, namely Nekemte Referral Hospital, Wollega University Referral Hospital, Bake Jama Health Center, and Cheleleki Health Center. Also, the town has more than fifteen private and NGO health facilities. From these health facilities in Nekemte Town, the two public hospitals, Nekemte Referral Hospital, and Wollega University Referral Hospital provides follow-up services for diabetic patients residing in Nekemte Town and its surrounding communities. The study was conducted in the two public hospitals providing follow-up services for diabetic patients [12].

### Source and study population

All diabetic patients on follow-up attending public hospitals in Nekemte Town were considered as source populations. All randomly selected diabetic patients (Type I and II) who were attending diabetic clinic of public hospitals in Nekemte Town during the study period was the study population. Eligible participants were known type 1 or 2 diabetic mellitus patients who visited the diabetic centers for follow-up and age older than 18 years were used for inclusion criteria and women with gestational diabetic mellitus was excluded.

### Sample size determination and sampling technique

A single population proportion formula was used to determine the sample size. The sample size was calculated by taking proportion of 50%. Marginal error between sample size and population parameter of 5% (d = 0.05), and 95% confidence level (Z = 1.96), and 10% non-response rate was considered. So, the final sample size was 422.

All known DM patients visited the study hospitals for follow-up were taken into consideration. Systematic random sampling was used to select the study participants. The sampling interval was calculated by dividing the total number of DM patients on follow-up as counted from the registries by the calculated sample size. Evidence from the study of hospitals shows there are 3200 DM patients on the follow-up in Nekemte Referral Hospital and 208 at Wollega University.

The sample interval (k) of the study is 8 (3408/422). The first candidate for the study was selected by simple random sampling from the first 8 (k) patients who arrived at the diabetic clinic on the first day of data collection and who met the eligibility criteria. The study subject selection continues in every eight intervals until the desired sample size is attained. The sample size was distributed between the study facilities based on proportion size of the study population.

## Data collection tool and techniques

The data was collected using a pretested structure questionnaire with closed-ended questions. The questionnaire is adapted from different relevant previous studies in the area [2, 4, 10] that adapted and modified to suit the current study. The questionnaire was prepared in English and translated to Afan Oromo by professional translators. The Afan Oromo questionnaires was then back translated to ensure that the original and translated questionnaires was check for similar in terms of content, clarity, and meaning. The back-translation to English was compared with the original questionnaire to ensure consistence of the questionnaires. The contents of the questionnaire were validated by pretest in the field. Data were collected using interviewer administered pretested questionnaire by nurse working in diabetic clinic of the health facility.

## Data quality control and management

To ensure the quality of data the following measures were undertaken. Validity of the questionnaire was maintained by pretested questionnaires 5% (21) of study population at Bako Hospital, Western Ethiopia. During the pre-test, the acceptability and applicability of the procedures and tools was evaluated. Training was given to data collectors on the objective of the study, data collection process and relevance of the study prior to data collection. The completed questionnaire was cross checked daily for inconsistencies. Throughout the course of the data collection, the data collectors were supervised at each site by the principal investigator. The data was checked for completeness on site and before data entry.

**Operational definition.** In this study, COVID-19 Vaccine hesitancy–A condition in which the study subject who refuse or fail to complete the vaccine despite availability of vaccination services; Diabetic Patients on Follow-Up–is known Diabetic Mellitus patients of any type who has a regular follow-up at diabetic clinic with the health care provider.

## Data management and analysis

The collected data was entered and cleaned using Epi-Data software 4.6 version. The cleaned data were analyzed using the Statistical Package for Social Sciences (SPSS) version 20.0 statically software. Descriptive statistics, including frequencies and proportions were used to summarize the study variables; quantitative variables are presented as mean and standard deviation. The crude odds ratio (COR) was obtained using a binary logistic regression model, with COVID-19 vaccine hesitancy as the dependent variable and baseline characteristics as independent variables. Variables with a P-values of $<0.05$ in the bi-variable logistic regression analysis were entered in the multivariable logistic regression analysis to control the possible effect of confounders. The adjusted odds ratio (AOR) with a 95% confidence interval was estimated to assess the strength of association, and a p value of $<0.05$ was used to declare independent variable to be the statistical significance determinate of COVID-19 hesitancy in the multivariable analysis.

## Ethical considerations

The ethical issue was approved by the Research and Ethical Review Committee of Rift Valley University with reference number RVU/AC/978/3/14. All administrative bodies communicated and obtained permission in a hierarchical manner. Then after verbal consent was sought by explaining the goals and methods of the study and their right to withdraw from participation at any time prior to the interview. One-page consent letter outlining the study's overall objective and confidentiality as no identifiers were used was attached to the cover page of each

questionnaire. The study had no procedure that would have an impact on the study subjects and the data would only be used for only research purpose.

## Result

### Socio-demography characteristics of study participants

In this study, 422 study participants took part in the study with a response rate of 100%. The mean age of the participants was 42.22 (SD±12.55) years. The age of the participants ranged from 18–75 years old. Among the study participants 158(37.4%), 103(24.4%) and 98(23.2%) were age between 30–39, 40–49 and > = 50 years old, respectively. Majority of the participants were male 281(66.6%), Oromo ethic group 360(85.3%), protestant religion 271(64.2%), and married 361(85.5%). Most (91.9%) of the study participants attended formal education. Three hundred twenty-three (76.5%) of the participants were urban dwellers, and about a quarter of respondents reported that their occupation was merchant 102 (24.2%) or government employee 116 (27.5%).

More than three third 169(40%) of the participants had earn an average monthly income of 1000–5000 Ethiopian Birr. More than half (51.9%) of participants had between 3 and 6 family size. The mean family size of a household was 4.52 (SD±2.254). Of the total, three hundred nighty-six (93.8%) study participants took part from Nekemte Referral Hospital, and the rest were from Wollega University Referral Hospital study institution (Table 1).

### Study participants' clinical characteristics

In the study, type-1 DM is the dominant 298 (70.6%) diabetic type. Around half (47.4%) of the study participants were diagnosed to be DM patients in the year between 5–10 years from the time of survey while 109(25%) within last 5 years, and the rest were over 10 years. One hundred three (24.4%) study participants had controlled glucose level (<126 gmd/dl) but around two third (75.6%) of the study participant had higher glucose level despite of they were undergoing the DM treatment follow. Most of the study participants describe their overall health condition as either average 141(33.4%) or good 214(50.7%) (Table 2).

### COVI-19 Vaccine awareness, source of information and practice

In this study, majority 392(92.9%) of study participants had awareness about the COVID-19 vaccine. Two-thirds of the study participants reported their primary source of information was Media (Television, Radio, Newspaper), and followed by healthcare provider 109(25.8%). More than half (55.2%) study participants had great deal awareness on COVID-19 vaccine preparation (Table 3).

Moreover, the study found, 250(59.2%) of study participants didn't receive any kind of vaccine in their lifetime. While 1347(82.2%) of study participants report, they support any vaccine. Vaccine hesitancy was higher 60(32.1%) among COVID-19 non- vaccinated study participants. The study also found majority 329 (78.0%) of the study participants were believes COVID-19 vaccine either definitely or probably reduce and protect complication from COVID-19 infection (Table 3).

Information related to COVID-9 expose and testing revealed, twenty-five (5.9%) study participants had their family had COVID-19 disease, and 17 (4.0%) family member died of COVID-19. Less the one-third 117 (27.7%) of study participants had tested for COVID-19 infection. Out of tested study participants seventeen (4%) were found to be positive. Around two-thirds (64.1%) of respondents also believe they were exposed to COVID-29 infection (Table 3).

**Table 1. Socio-demography characteristics of the diabetic patients attending public hospitals in Nekemte Town, East Wollega Zone, Western Ethiopia, 2023.**

| Variables | | Vaccine hesitancy | | Total, N (%) |
|---|---|---|---|---|
| | | Yes, N (%) | No, N (%) | |
| **Family size** | <3 | 25(18.1) | 113(81.9) | 138(32.7) |
| | 3–6 | 34(15.5) | 185(84.5) | 219(51.9) |
| | >6 | 5(7.7) | 60(92.3) | 65(15.4) |
| **Sex** | Male | 42(14.9) | 239(85.1) | 281(66.6) |
| | Female | 22(15.6) | 119(84.4) | 141(33.4) |
| **Age** | 18–29 | 15(23.8) | 48(76.2) | 63(14.9) |
| | 30–39 | 21(13.3) | 137(86.7) | 158(37.4) |
| | 40–49 | 23(22.3) | 80(77.7) | 103(24.4) |
| | > = 50 | 5(5.1) | 93(94.9) | 98(23.2) |
| **Ethnicity** | Oromo | 53(14.7) | 307(85.3) | 360(85.3) |
| | Amhara | 6(12.8) | 41(87.2) | 47(11.1) |
| | Other* | 5(33.3) | 10(66.7) | 15(3.6) |
| **Marital status** | Single | 10(22.7) | 34(77.3) | 44(10.4) |
| | Married | 50(13.9) | 311(86.1) | 361(85.5) |
| | Window | 1(9.1) | 10(90.9) | 11(2.6) |
| | Divorced | 3(50.0) | 3(50.0) | 6(1.4) |
| **Education** | No formal education | 7(20.6) | 27(79.4) | 34(8.1) |
| | Elementary | 11(12.2) | 79(87.8) | 90(21.3) |
| | Secondary/high school | 19(14.2) | 115(85.8) | 134(31.8) |
| | College diploma | 7(13.5) | 45(86.5) | 52(12.3) |
| | University level degree | 20(17.9) | 92(82.1) | 112(26.5) |
| **Religion** | Protestant | 50(18.5) | 221(81.5) | 271(64.2) |
| | Orthodox | 10(8.1) | 114(91.9) | 124(29.4) |
| | Muslim | 4(14.8) | 23(85.2) | 27(6.4) |
| **Occupation** | Unemployed | 3(8.6) | 32(91.4) | 35(8.3) |
| | Farmer | 9(11.5) | 69(88.5) | 78(18.5) |
| | Student | 6(25.0) | 18(75.0) | 24(5.7) |
| | Marchant | 14(13.7) | 88(86.3) | 102(24.2) |
| | Miner | 0(0.0) | 2(100.0) | 2(0.5 |
| | Governmental employee | 12(10.3) | 104(89.7) | 116(27.5) |
| | Religious leader | 12(75.0) | 4(25.0) | 16(3.8) |
| | Housewife | 3(30.0) | 7(70.0) | 10(2.4) |
| | Daily laborer | 5(12.8) | 34(87.2) | 39(9.2) |
| **Income per monthly (Ethiopia birr)** | <1000 | 22(20.2) | 87(79.8) | 109(25.8) |
| | 1000–5000 | 28(16.6) | 141(83.4) | 169(40.0) |
| | >5000 | 14(9.7) | 130(90.3) | 144(34.1) |
| **Address** | Urban | 49(15.2) | 274(84.8) | 323(76.5) |
| | Rural | 15(15.2) | 84(84.8) | 99(23.5) |
| **Study institution** | NRH | 62(15.7) | 334(84.3) | 396(93.8) |
| | WURH | 2(7.7) | 24(92.3) | 26(6.2) |

Key: N = Frequency, % = Percentage

* = Gurage, Tigre, NRH = Nekemte Referral Hospital, WURH = Wollega University Referral Hospital

**Table 2. Clinical characteristics of diabetic patients attending public hospitals in Nekemte Town, East Wollega Zone, Western Ethiopia, 2023.**

| Variables | | Vaccine hesitancy | | Total, N (%) |
|---|---|---|---|---|
| | | Yes | No | |
| DM diseased year | <5 | 19(17.4) | 90(82.6) | 109(25.8) |
| | 5–10 | 28(14.0) | 172(86.0) | 200(47.4) |
| | >10 | 17(15.0) | 96(85.0) | 113(26.8) |
| DM type | Type-1 | 49(16.4) | 249(83.6) | 298(70.6) |
| | Type-2 | 15(12.1) | 109(87.9) | 124(29.4) |
| Other chronic diseases | yes | 19(17.9) | 87(82.1) | 106(25.1) |
| | No | 45(14.2) | 271(85.8) | 316(74.9) |
| Fasting glucose level (**mg/dl**) | <126 | 10(9.3) | 93(90.3) | 103(24.4) |
| | 126–200 | 36(17.2) | 173(82.8) | 209(49.5) |
| | >200 | 18(16.4) | 92(83.6) | 110(26.1) |
| Overall health | Very poor | 0(0.0) | 16(100.0) | 16(3.8) |
| | Poor | 6(28.6) | 15(71.4 | 21(5.0) |
| | Average | 28(19.9) | 113(80.1) | 141(33.4) |
| | Good | 27(12.6) | 187(87.4) | 214(50.7) |
| | Very good | 3(10.0) | 27(90.0) | 30(7.1) |

Key: N = Frequency, % = Percentage, DM = Diabetes Mellitus, mg = **milligrams**, dl = deciliter

## COVID-19 vaccine hesitancy rate and its reasons

The overall magnitude of COVID-19 vaccine hesitancy rate was 15.2% (95% CI: 11.6–18.7) (Fig 1). The vaccine hesitancy rate was 12.10% and 16.40% among type -2 and type 1 diabetic patients, respectively. The top listed reasons for the COVID-19 vaccine hesitancy were negative information about the vaccine (32.90%), lack of enough information (21.80%), concern about vaccine side effects (19.40%), didn't believe the vaccine work and/or effective (15.4%), didn't believe COVID-19 is health problem (14.70%), and not at risk of contracting COVID-19 (12.80%) (Fig 2).

## Predictors of vaccine hesitancy

The study revealed that respondents in the age of 40–49 years old had 4.52 more likely to hesitate COVID-19 vaccination compared to those respondents aged greater than 50 years old (AOR = 4.52(1.04–19.66)). The study participants who had COVID-19 vaccine awareness were 0.029 times less likely to hesitate COVID-19 vaccine (AOR = 0.029(0.001–0.857)) compared to their counterpart. The odd of having previous COVID-19 vaccine were 0.134 less compared to their counterpart to vaccine hesitancy (AOR = 0.134(0.035–0.507)). Also, the study participants who had a great deal (AOR = 0.028(0.002–0.523)) and a fear (AOR = 0.046(0.003–0.791)) trust on vaccine preparation or development were 0.03 and 0.05 times less likely to hesitate to the vaccine comparing to those respondents who had no trust at all. Moreover, respondents who strongly agree and/or agree (AOR = 0.03(0.006–0.17)) and neither agree nor disagree (AOR = 0.07(0.02–0.30)) to the perceived COVID-19 infection exposure were 0.03 and 0.07 times less likely to hesitate the vaccine comparing to the stronger disagree and or disagree (Table 4).

## Discussion

Vaccination against COVID-19 can significantly reduce the risk of COVID-19 diseases, complications, and hospitalization in patients with chronic diseases including diabetes. The

**Table 3. Awareness, source of information and practice on COVID-19 vaccine hesitancy among diabetic patients attending public hospitals in Nekemte Town, East Wollega Zone, Western Ethiopia, 2023.**

| Variable | | Vaccine hesitancy | | Total, N (%) |
|---|---|---|---|---|
| | | Yes | No | |
| COVID-19 Vaccine awareness | | | | |
| | Yes | 54(13.8) | 338(86.2) | 392(92.9) |
| | No | 10(33.3) | 20(66.7) | 30(7.1) |
| Source of information | | | | |
| | Media (TV, Radio, Newspaper) | 51(17.4) | 242(82.6) | 293(69.4) |
| | Healthcare provider | 8(8.7) | 101(92.7) | 109(25.8) |
| | Friends & family member | 3(30.0) | 7(70.0) | 10(2.4) |
| | Religious leaders | 1(20.0) | 4(80.0) | 5(1.2) |
| | Social media | 1(20.0) | 4(80.0) | 5(1.2) |
| Family members have COVID-19 | | | | |
| | No | 59(14.9) | 338(85.1) | 397(94.1) |
| | Yes | 5(20.0) | 20(80.0) | 25(5.9) |
| Tested for COVID-19 | | | | |
| | No | 51(16.7) | 254(83.3) | 305(72.3) |
| | Yes | 13(11.1) | 104(88.9) | 117(27.7) |
| Tested positive for COVID-19 | | | | |
| | No | 64(15.8) | 341(84.2) | 405(96.0) |
| | Yes | 0(0.0) | 17(100.0) | 17(4.0) |
| Close contact with COVID-19 patient | | | | |
| | No | 59(15.0) | 335(85.0) | 394(93.4) |
| | Yes | 5(17.9) | 23(82.1) | 28(6.6) |
| Family died of COVID-19 | | | | |
| | No | 61(15.1) | 344(84.9) | 405(96.0) |
| | Yes | 3(17.6) | 14(82.4) | 17(4.0) |
| Received any vaccine in lifetime | | | | |
| | No | 35(14.0) | 215(86.0) | 250(59.2) |
| | Yes | 26(16.1) | 135(83.9) | 161(38.2) |
| | Unknown | 3(27.3) | 8(72.7) | 11(2.6) |
| No support any vaccine | | | | |
| | Yes | 26(40.6) | 38(59.4) | 64(15.2) |
| | No | 34(9.8) | 313(90.2) | 347(82.2) |
| | Unknown | 4(36.4) | 7(63.6) | 11(2.6) |
| Vaccinated for COVID-19 | | | | |
| | Yes | 4(1.7) | 231(98.3) | 235(55.7) |
| | No | 60(32.1) | 127(67.9) | 187(44.3) |
| Exposed to COVID-19 infection | | | | |
| | Strong agree | 1(0.6) | 171(99.4) | 172(40.8) |
| | Agree | 4(7.1) | 52(92.9) | 56(13.3) |
| | Neither agree nor disagree | 36(21.7) | 130(78.3) | 166(39.3) |
| | Disagree | 14(82.4) | 3(17.6) | 17(4.0) |
| | Strongly disagree | 9(81.8) | 2(18.2) | 11(2.6) |
| Awareness on vaccine preparation | | | | |
| | A great deal | 6(2.6) | 227(97.4) | 233(55.2) |
| | A fear amount | 7(8.6) | 74(91.4) | 81(19.2) |
| | No too much | 46(45.1) | 56(54.9) | 102(24.2) |

(*Continued*)

**Table 3.** (Continued)

| Variable | | Vaccine hesitancy | | Total, |
|---|---|---|---|---|
| | | Yes | No | N (%) |
| | None at all | 5(83.3) | 1(16.7) | 6(1.4) |
| COVID-19 vaccine reduces and protect from complication | | | | |
| | Definitely | 2(0.9) | 217(99.1) | 219(51.9) |
| | Probably | 7(10.9) | 103(28.8) | 110(26.1) |
| | Unsure | 22(40.0) | 33(60.0) | 55(13) |
| | Probably not | 23(35.9) | 3(0.8) | 26(6.2) |
| | Definitely not | 10(83.3) | 2(16.7) | 12(2.8) |

Key: N = Frequency, % = Percentage

existing evidence showed that to stop the spread of the COVID-19 pandemic and to develop herd immunity, 60–70% of society should be vaccinated [13]. Therefore, the highest acceptance of the COVID-19 vaccine has a greater role to control the worldwide COVID-19 pandemic. However, its effectiveness is challenged by vaccine hesitancy.

In this study the overall magnitude of COVID-19 vaccine hesitancy rate was 15.2% (95% CI: 11.6–18.7). This is in line with the study conducted in Italian (14.2%) [14], Woldia, Ethiopia (17.4%) [15], Uganda (15.5%) [16], Malawi (17.3%) [16] but lower than study conducted at Sub-Saharan Africa (26.0%) [17], China (56.4%) [18] and global (24.9%) [19], On the other hand, the magnitude was lower than from findings among the general population in Ethiopia (42.2%) [20]. The discrepancy in these data may be due to sociodemographic characteristics. These difference in vaccine hesitancy rate is also partly explained with COVID-19 vaccine hesitancy is not stable and changing with time. This is well presented in the cohort study done by Aaron *et al.*, 2021 [21, 22] that where COVID-19 vaccine hesitancy was decreased between late 2020 and early 2021, with nearly one-third (32%) of persons who were initially hesitant being

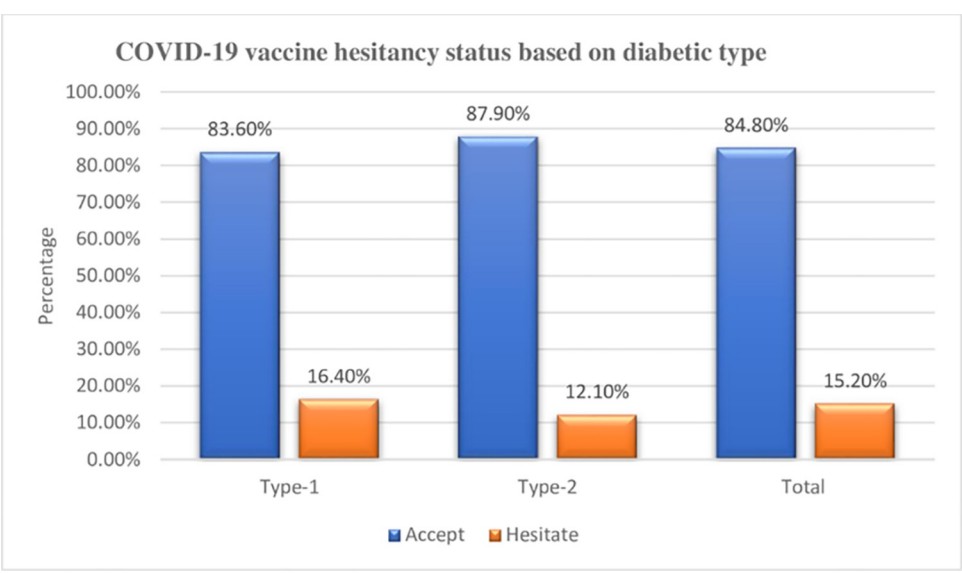

**Fig 1. COVID-19 vaccine hesitancy rate among of diabetic patients attending public hospitals in Nekemte Town, East Wollega Zone, Western Ethiopia, 2023.**

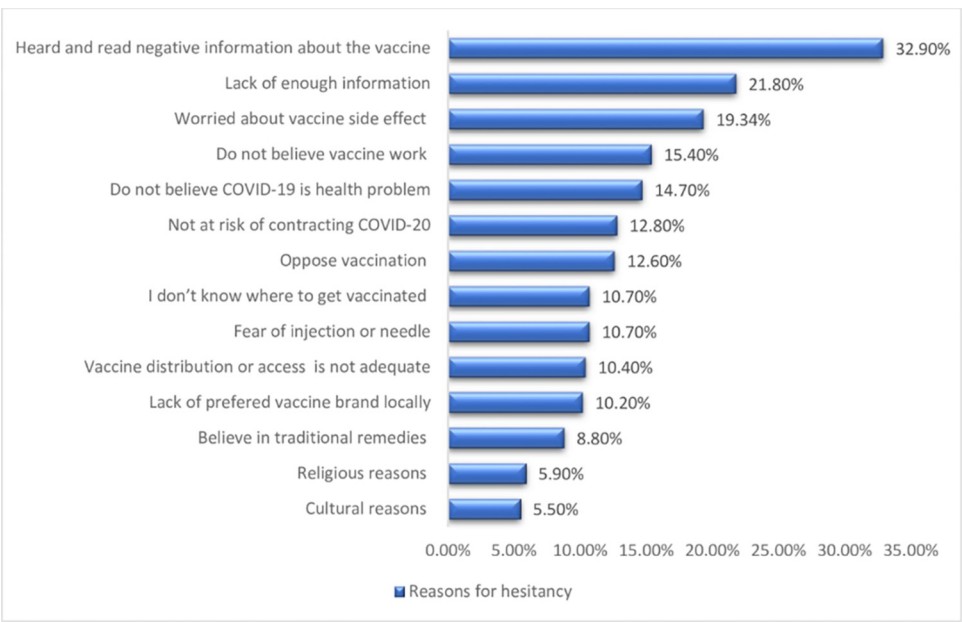

**Fig 2. Reasons for hesitancy of the COVID-19 vaccine among of diabetic patients attending public hospitals in Nekemte Town, East Wollega Zone, Western Ethiopia, 2023.**

vaccinated at follow-up and more than one-third (37%) transitioning from vaccine hesitant into vaccine willing.

The reasons for COVID-19 vaccine acceptance and hesitancy remain complex. As new COVID-19 variants emerge, adding further complexity [23], new vaccines come to the market, it will be important to maintain a delicate balance in communicating what is known and acknowledging the uncertainties that remain. In this study, a top reason for a study participant to hesitate to get COVID-19 vaccine was negative information about the vaccine and followed with lack of enough information and fear of vaccine side effect. In support of these, the study found the participants who had vaccine awareness had 0.03 less odd to hesitate. This data confirms evidence already documented on other studies done among diabetic patients and other high-risk populations [24–26]. The evidence further notes the need to avoid spreading falsehoods or using language that could be misinterpreted and could thereby potentially add to vaccine hesitancy as well as increases awareness to increase vaccine uptake.

Surprisingly, the study revealed having a great deal and/or a fear amount of awareness on the vaccine preparation would lower participants vaccine hesitancy. This is in support of the existing literature [25, 26]. This attribute to the facts that knowledge on the vaccine preparation and studying finding in the clinical trial help high risk individuals like diabetic patients to understand the value of the vaccine, possible side effect and to avoid misinformation.

The association between the COVID-19 vaccine hesitancy and socio-demographic variables of the study participants supports the scale's construct validity. The study results indicated that age between 40–49 was associated with higher likely to vaccine hesitate to compare to older, a finding which aligns with findings from previous studies that found the intention to get vaccinated increases with age [27, 28].

In present study found previous COVID-19 vaccination is one of the main independent predictors for COVID-19 vaccine hesitancy where study participants who have previous vaccination history had hesitancy to COVID-19 vaccination with the odd of 0.13. This is in line with existing literature [29, 30]. Perceived risk could also be the predictors of COVID-19

**Table 4. Binary logistic regression analysis result for predictor variables with COVID-19 vaccine hesitancy among diabetic patients attending public hospital in Nekemte Town, East Wollega Zone, Western Ethiopia, 2023.**

| Variables | | Hesitate, N (%) | COR (95%CI) | P-value | AOR (95%CI) | P-value |
|---|---|---|---|---|---|---|
| Age | 18–29 | 15(23.8) | 5.812(1.99–16.95) | **0.001*** | 2.002(0.36–11.17) | 0.43 |
| | 30–39 | 21(13.3) | 2.851(1.04–7.83) | **0.042*** | 1.842(0.443–7.663) | 0.401 |
| | 40–49 | 23(22.3) | 5.35(1.94–14.72) | **0.001*** | 4.516(1.04–19.66) | **0.045*** |
| | > = 50 | 5(5.1) | 1 | | 1 | |
| Occupation | Unemployed | 3(8.6) | 0.64(0 .14–2.89) | 0.559 | 0.19(0.02–1.81) | 0.148 |
| | Farmer | 9(11.5) | 0.89(0.28–2.852) | 0.840 | 0.25(0.04–1.75) | 0.162 |
| | Student | 6(25.0) | 2.267(0.61–8.46) | 0.223 | 0.87(0.09–8.04) | 0.902 |
| | Marchant/miner | 14(13.5) | 1.06(0.35–3.16) | 0.888 | 0.37(0.06–2.49) | 0.310 |
| | Governmental employee | 12(10.3) | 0.79(0.26–2.387) | 0.669 | 0.83(0.13–5.25) | 0.843 |
| | Religious leader | 12(75.0) | 20.40(0.69–88.75) | **<0.001*** | 5.93(0.50–70.22) | 0.158 |
| | Housewife | 3(30.0) | 2.91(0.56–15.12) | 0.203 | 2.22(0.17–28.34) | 0.540 |
| | Daily laborer | 5(12.8) | 1 | | 1 | |
| Income per monthly (Ethiopia birr) | <1000 | 22(20.2) | 2.35(1.14–4.84) | **0.021*** | 1.373(0.33–5.73) | 0.663 |
| | 1000–5000 | 28(16.6) | 1.84(0.930–3.66) | 0.080 | 1.65(0.48–5.67) | 0.427 |
| | >5000 | 14(9.7) | 1 | | 1 | |
| Source of information | Media (TV, Radio, Newspaper) | 51(17.4) | 0.58(0.18–1.89) | 0.366 | 0.71(0.10–5.27) | 0.74 |
| | Healthcare provider | 8(8.7) | 0.22(0.06–0.842) | **0.027*** | 0.81(0.10–6.83) | 0.85 |
| | Other ** | 4(26.7) | 1 | | 1 | |
| Having COVID-19 Vaccine awareness | Yes | 54(13.8) | 0.32(0.14–0.72) | **0.006*** | 0.029(0.00–0.86) | **0.040*** |
| | No | 10(33.3) | 1 | | | |
| No support any vaccine | Yes | 26(40.6) | 1.197(0.32–4.51) | 0.790 | 1.33(0.24–7.30) | 0.747 |
| | No | 34(9.8) | 0.19(0.05–0.68) | **0.011*** | 0.364(0.07–1.87) | 0.226 |
| | Unsure | 4(36.4%) | 1 | | | |
| Vaccinated for COVID-19 | Yes | 4(1.7) | 0.04(0.01–0.10) | **<0.001*** | 0.13(0.035–0.51) | **0.003*** |
| | No | 60(32.1) | 1 | | | |
| Exposed to COVID-19 infection | Strong agree/agree | 5 (2.2) | 0.01(0.00–0.02) | **<0.001*** | 0.031(0.01–0.17) | **<0.001*** |
| | Neither agree nor disagree | 36 (21.7) | 0.06(0.02–0.17) | **<0.001*** | 0.071(0.02–0.30) | **<0.001*** |
| | Disagree/ Strongly disagree | 23 (82.1) | | | | |
| Trust on vaccine preparation process | A great deal | 6(2.6) | 0.01(0.00–0.05) | **<0.001*** | 0.03(0.00–0.52) | **0.017*** |
| | A fear amount | 7(8.6) | 0.019(0.00–0.19) | **0.001*** | 0.05(0.00–0.79) | **0.034*** |
| | No too much | 46(45.1) | 0.164(0.02–1.46) | 0.105 | 0.14(0.01–1.99) | 0.147 |
| | None at all | 5(83.3) | 1 | | | |

N = Frequency, % = Percentage

* = Statistically significant, COR = Crude odd ratio, 95% CI = 95% confident interval, 1 = reference, and AOR = Adjusted odd ratio

** = Friends & family member, religious leaders, social media (FB, Whatup, Twitter etc)

vaccine acceptance in the existing literature [28] which is consistent with the results of this study, the higher perceived risk of COVID-19 infection was associated with lower odd of vaccine. This is attributed to participants perceived risk level is a determinate to accept or hesitate a COVID-19 vaccine as one of method to avoid the risk of their own health or the health of their loved ones. The possible explanation for vaccine hesitancy towards COVID-19 vaccination could be individuals with a history of vaccinations may have experienced side effects from past vaccinations, leading them to be hesitant about receiving the COVID-19 vaccine. Past experiences with vaccine side effects can influence an individual's attitudes and perceptions

towards vaccination in general, and these attitudes can persist even when the benefits of vaccination outweigh the risks.

## Strength and limitation of the study

A strength of this study is that it is the first to investigate predictor variables with COVID-19 vaccine hesitancy in diabetic patients on follow-up in Ethiopia. Hence, the study gives further insights on the magnitude of COVID-19 vaccines hesitancy and its predictor variables among high-risk diabetic patients in the country. However, our study also had some limitations. First, this study design to measured COVD-19 vaccines hesitancy at a certain point in time that is potentially prone to change with the vaccine availability and level of the problem in the country. Lastly, as the study was a cross-sectional survey, the causal relationship between predictors and outcome variables could not be determined.

## Conclusion and recommendation

The study confirms that COVID-19 vaccine hesitancy among diabetic patient was relatively low. The hesitancy of the COVID-19 vaccination uptake among diabetes patients was independently influenced by age, vaccine awareness, COVID-19 vaccination history, awareness on vaccine preparation and exposure status to COVID-19 infection. Hence, stakeholders have to focus on efforts to translating these high levels of vaccine acceptance into actual uptake, through ensuing vaccine availability and accessibility vaccine to for a high-risk diabetes patient. Policy makers should design policies to integrate COVID-19 health education in the ongoing diabetic management in the health care system to avoid the ongoing misinformation and conspiracies on the diseases. Behavioral change communication should be promoted about the value of vaccine, the safety, and level of protection of the vaccine for individual with diabetic. In a future study, we recommended larger sample size that employ mixed research methods both qualitative and quantitative approaches to fully capture the COVID-19 vaccines hesitancy in Ethiopia.

## Supporting information

**S1 Data.**
(SAV)

## Acknowledgments

The authors would like to acknowledge the Nekemte Referral Hospital and Wollega University Referral Hospital administration for their unreserved support during data collection. The author also grateful for all data collectors, and respondents without whom this research would not have been realized.

## Author Contributions

**Conceptualization:** Aberash Olani Kuta, Nagasa Dida.

**Data curation:** Aberash Olani Kuta, Nagasa Dida.

**Formal analysis:** Aberash Olani Kuta, Nagasa Dida.

**Investigation:** Aberash Olani Kuta, Nagasa Dida.

**Methodology:** Aberash Olani Kuta, Nagasa Dida.

**Project administration:** Aberash Olani Kuta, Nagasa Dida.

**Resources:** Aberash Olani Kuta.

**Software:** Aberash Olani Kuta, Nagasa Dida.

**Supervision:** Nagasa Dida.

**Validation:** Nagasa Dida.

**Visualization:** Nagasa Dida.

**Writing – original draft:** Aberash Olani Kuta, Nagasa Dida.

**Writing – review & editing:** Aberash Olani Kuta, Nagasa Dida.

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
