## [Decision Letter · Decision Letter 0]

20 Nov 2023

PONE-D-23-22981Covid-19 Vaccine Hesitancy and Predictor Factors among Diabetic Patients on Follow-Up at Public Hospitals in Nekemte Town, Western EthiopiaPLOS ONE

Dear Dr. Dida,

Thank you for submitting your manuscript to PLOS ONE. After careful consideration, we feel that it has merit but does not fully meet PLOS ONE’s publication criteria as it currently stands. Therefore, we invite you to submit a revised version of the manuscript that addresses the points raised during the review process.

**ACADEMIC EDITOR: **Try to address the following points in addition to reviewers comments:The paper has major grammar and language flaws that needs intense revisions. I strongly recommend this paper be edited by subject language experts.  Abstract: This section should be concise but informative. Describe aim of the study, and make clearer the sampling strategy, and analysis model and approaches. For results (AOR), limit the decimal points to 2 decimal points and explore the reasons for wider confidence interval. Correct the data availability statement. PLOS ONE journal requires authors to supply data used in this paper or clearly indicate the site where the data can be easily accessed. If data are not shared at this time, authors should explain why they are not supplying research data. Rewrite the research ethics statement (the flow of idea is distorted) and describe how the confidentiality of respondents information is maintained. Change the in-text citation format to square brackets (such as: [1])Introduction: Line # 47-49: How you are sure that existing COVID-19 control measures(including vaccine) had no impact in stopping disease transmission, hospitalization and death? Provide an evidence that supports this. In the first use, describe abbreviations in full terms (e.g., DM, line #57). Try to provide some contextual information about COVID-19 in Ethiopia, including the date it is notified; the burden it had and the different control measures adapted and implemented. Supplement information about the prevailing factors of vaccine hesitancy in Ethiopia. Try to also to review studies that conducted focusing on high-risk patients or population groups. Methods: Describe the contents of the questionnaire, and supply the English version of the tool as supplementary information. How the final regression model fitness was assessed. Have you did multicollinearity test? if so, what is the result?  Why you used P<0.05 in the bivariate regression? Operationalize some terms such as: Overall health. Results: Correct the interpretations of the findings and make it easily understandable to readers (Line # 219 and afterwards) Please submit your revised manuscript by Jan 04 2024 11:59PM. If you will need more time than this to complete your revisions, please reply to this message or contact the journal office at plosone@plos.org. Please include the following items when submitting your revised manuscript:A rebuttal letter that responds to each point raised by the academic editor and reviewer(s). You should upload this letter as a separate file labeled 'Response to Reviewers'.A marked-up copy of your manuscript that highlights changes made to the original version. You should upload this as a separate file labeled 'Revised Manuscript with Track Changes'.An unmarked version of your revised paper without tracked changes. You should upload this as a separate file labeled 'Manuscript'.

We look forward to receiving your revised manuscript.

Kind regards,

Dawit Wolde Daka

Academic Editor

PLOS ONE

- 10.1136/bmjopen-2021-052432

- http:// dx.doi.org/10.4314/ejhs.v32i6.2

- https://doi.org/10.1038/s41591-021-01459-7

- https://blogs.imperial.ac.uk/medical-centre/author/amajeed/

- https://health.gov.on.ca/en/pro/ministry/research/docs/raeb_eu_78.pdf

In your revision ensure you cite all your sources (including your own works), and quote or rephrase any duplicated text outside the methods section. Further consideration is dependent on these concerns being addressed.

Reviewers' comments:

Reviewer's Responses to Questions

**Comments to the Author**

1. Is the manuscript technically sound, and do the data support the conclusions?

Reviewer #1: Yes

Reviewer #2: Yes

2. Has the statistical analysis been performed appropriately and rigorously? 

Reviewer #1: Yes

Reviewer #2: Yes

3. Have the authors made all data underlying the findings in their manuscript fully available?

Reviewer #1: Yes

Reviewer #2: Yes

4. Is the manuscript presented in an intelligible fashion and written in standard English?

Reviewer #1: No

Reviewer #2: No

5. Review Comments to the Author

Reviewer #1: Dear Editor, thank you for allowing me to review this manuscript. This manuscript need major revision. The following are my comments and suggestion.

1. Abstract sections line 23 to 24, what does it mean …………………………by trained nursing working at study institutions (nurses and nursing are different).

2. Abstract section (method subsection)- more information (like the model used, p- value) to analyse should be added

3. Abstract line 31 (Adjusted Odd Ratio [AOR]

4. Abstract line 32: having a great deal of what?

5. Abstract line 33: a fear amount (AOR= 0.046(0.003-0.791)) of awareness on the vaccine preparation ? (not clear for me)

6. Abstract line 34-36: exposure to COVID-19 infection status as strongly agree/agree (AOR = 0.031(0.006-0.171))and neither agree nor disagree (AOR = 36 0.071(0.017-0.295)). How did you specifically see the association between strongly agree/agree and neither agree nor disagree with vaccine hesitancy to the question of ‘’COVID-19 infection status’’? (Likert scale have its own way analysis method).

7. Abstract section-conclusion subsection line 37: The study confirms low rate of COVID-19 vaccine hesitancy……………the term ‘’confirms’’ and ‘’rate’’ should be modified

8. Abstract line 39-42: The relevant agency should focus on efforts to translating these high levels of vaccine acceptance into actual uptake, through targeting identifying determinate factors and vaccine availability for a high-risk diabetes patient. Your result and conclusion contradict each other (low rate of vaccine hesitancy and high level of vaccine acceptance). Grammatically wrong sentences (……..through targeting identifying…………)

9. Most references cited at the end of paragraph throughout the documents. It is recommended to cite after each sentences

10. Introduction line 55 (cite references after ……..COVID-19)

11. Introduction line 58 (cite references after ……SARS-CoV-2 infection)

12. Introduction line 59 (….vaccine hesitancy varying from 76.4% to 3.0%). Better if changed to ….varying from 3.0% to 76.4%

13. Introduction line 67 ( cite references after …….sociodemographic contexts)

14. Introduction line 71- you only cite one reference (8) for two study findings (81.6% in South Africa to 65.2% in Nigeria). How it could be?

15. Again introduction line 71-72 This sentence ‘’In Ethiopia, the hesitancy rate range between 19.1%- 60.3%% were documented in the studies conducted in different part of the country’’ should be cited for each both findings.

16. Introduction line 77-76- repetitions (….among the among high-risk population….)

17. Introduction line 77-80- the last sentences is too long sentence and difficult to catch up the main idea of the sentences. Better if break it down to make mor meaningful and understandable.

18. Line 96-97- what to mean? ‘’All selected diabetic patients (Type I and II) who were attending diabetic clinic of public hospitals in Nekemte Town during the study period’’

19. Exclusion criteria?

20. Line 102-103- why did you use 50% for sample size calculation? You wrote that in introduction section numerous studies reported the hesitancy rate range between 19.1%-60.3%%.

21. Line 105- ‘’A total of 422 patients with known DM participated in the study’’ this sentences should be under result sections (It should be as follows; So, the final sample size was 422).

22. Line 108- from when to when you counted the total number of DM patients on follow-up from the registries?

23. Line 117-119- ‘’ The 118 questionnaire is adapted from different relevant previous studies in the area (2, 4, 10) that adapted 5 119 and modified to suit the current study’’. Should be rewritten

24. Line 124-125 ‘’The questionnaire was administered face to 125 face by trained nursing working in the study facility diabetic clinic or Wards’’. Should be re-written

25. Line 129- add the actual number on which you conducted pre-test

26. Line 143-144- ‘’Variables with a P-values of <0.05 in the bi-variable logistic regression analysis were entered in the multivariable logistic regression analysis to control the possible effect of confounders’’. What is your base to use p-value <0.05 to select your candidate variables in bi-variable logistic regression?

27. What are about multicollinearity and model fitness test?

28. General comments on result sections; Almost you doubled your result section, (you wrote in texts and again you put those variables in the table in almost all of your result sections. If you explain it in text or by paragraph, you are not expected to put those variable in the table form and vice versa.

29. Line 184-185- how did you measure whether the overall health condition of study participants as very poor, poor, average, good and very good?

30. Line 218 to 229- I didn’t like the way the factors were written in all. I strongly recommend the authors to talk and consult the staticians (biostaticians and epidemiologist).

31. Line 230- Table 4: add multivariable logistic regression also on the title

32. In the table: not ‘’income per monthly’’ rather ‘’Monthly income’’

33. Discussion line 270-282- not well discussed.

Reviewer #2: Tittle:

• Change the title to "COVID-19 Vaccine Hesitancy and its Predictors among Diabetic Patients on Follow-Up at Public Hospitals in Nekemte Town, Western Ethiopia."

Abstract

• Line 25, “Statically” to “statistical”.

Introduction

• The gaps weren't stated well, especially in relation to patients with diabetes, even though the introduction was written well. Three key considerations need to be made in this instance;

Principles and practices

Gaps, which discuss what was and wasn't known

Filling the gaps

Methods and materials

• Line 96-97, “All selected diabetic patients (Type I and II) who were attending diabetic clinic of public hospitals in Nekemte Town during the study period”. What this statement indicates? Who are your study populations?

• Why you have considered P=50% for sample size calculation? Justification?

• How you have calculated K (8) value for both hospital and how it could be? Clarify the proportional allocation of the sample to each study hospitals.

• Line 118 ,rephrase ”The questionnaire is adapted from different relevant previous studies in the area (2, 4, 10) that adapted and modified to suit the current study”

• Line 120 “Afan Oromo (local language)” to “Afan Oromo” omitting local language

• Have you actually verified the questionnaire's validity? What was your proof, if any?

• Be consistent while using the terms like predictors, factors and determinant.

Result

• The phrase "In this study" appears multiple times in lines 161,169, 172 and etc. in the manuscript. Put it in a different way.

• Line 164 “>=50” to “>50” and also in table 1

• Line 174 Clinical Characteristics study participants to “Study participants' clinical characteristics”

• Clearly define any terms. As stated in Table 2, the DM diseased year, DM type, and overall health

• Line 202, less the one-third? Means?

• In table 3, you have stated that as” Tested positive for COVID-19 - --17(4.0%) and Family died of COVID-19---17(4.0%).What this indicate??

• Table 3 reported that 235 individuals (55.7%) had received the COVID-19 vaccine; however, the overall rate of vaccine hesitancy was 15.2%. This suggests that 84.8% of people do not have vaccine hesitancy. Conversely, what can we say about those 29.1% in this regard?

• Line 220, “The study participants who have vaccine awareness had 0.029 less odd to hesitate (AOR= 0.029(0.001222 0.857)”.Rephrase it and how you have described the odd ratio was not clear. Please apply this concern to all.

Discussion

• As per your objectives, the discussion section ought to center around the relevant findings that emerged from the result section. Modifications are necessary to the way it has been discussed.

Conclusion

• I felt that your findings and recommendation in this section were in conflict with one another. Your recommendation should therefore be based on your relevant findings and should be appropriate.

Generally

• The manuscript needs to be re-narrated in clear, concise English, with consistency and coherence in the use of terminology.

6. PLOS authors have the option to publish the peer review history of their article (what does this mean?). If published, this will include your full peer review and any attached files.

Reviewer #1: No

Reviewer #2: No

---

## [Author Response · Author response to Decision Letter 0]

23 Jan 2024

Thanks the reviewers for you very nice comments.

I have attached response to reviewers query as independent files named response to reviewers.

---

## [Decision Letter · Decision Letter 1]

27 May 2024

Covid-19 Vaccine Hesitancy and Its Predictors among Diabetic Patients on Follow-Up at Public Hospitals in Nekemte Town, Western Ethiopia

PONE-D-23-22981R1

Dear Dr. Dida,

We’re pleased to inform you that your manuscript has been judged scientifically suitable for publication and will be formally accepted for publication once it meets all outstanding technical requirements.

Kind regards,

Dawit Wolde Daka

Academic Editor

PLOS ONE

Additional Editor Comments (optional):

Reviewers' comments:

Reviewer's Responses to Questions

**Comments to the Author**

1. If the authors have adequately addressed your comments raised in a previous round of review and you feel that this manuscript is now acceptable for publication, you may indicate that here to bypass the “Comments to the Author” section, enter your conflict of interest statement in the “Confidential to Editor” section, and submit your "Accept" recommendation.

Reviewer #1: All comments have been addressed

2. Is the manuscript technically sound, and do the data support the conclusions?

Reviewer #1: Yes

3. Has the statistical analysis been performed appropriately and rigorously? 

Reviewer #1: Yes

4. Have the authors made all data underlying the findings in their manuscript fully available?

Reviewer #1: Yes

5. Is the manuscript presented in an intelligible fashion and written in standard English?

Reviewer #1: Yes

6. Review Comments to the Author

Reviewer #1: No comments from my side and all comments were addressed by the authors. I want to say congratulations to all authors of this wonderful scientific paper.

7. PLOS authors have the option to publish the peer review history of their article (what does this mean?). If published, this will include your full peer review and any attached files.

Reviewer #1: **Yes: **Addisu Sertsu

---

## [Editor Report · Acceptance letter]

30 May 2024

PONE-D-23-22981R1 

PLOS ONE

Dear Dr. Dida, 

I'm pleased to inform you that your manuscript has been deemed suitable for publication in PLOS ONE. Congratulations! Your manuscript is now being handed over to our production team.

Kind regards, 

on behalf of

Mr Dawit Wolde Daka 

Academic Editor

PLOS ONE